# Active for Life after Cancer: Association of Physical Activity with Cancer Patients’ Interpersonal Competence, Quality of Life, and Survival Beliefs

**DOI:** 10.3390/bs13060449

**Published:** 2023-05-29

**Authors:** Ying Liu, Longjun Jing, Yang Liu, Huilin Wang, Tinggang Yuan, Jingyu Yang

**Affiliations:** 1School of Physical Education, Hunan University of Science and Technology, Xiangtan 411201, China; 2China Institute of Sports Science, Beijing 100061, China; yuantinggang@ciss.cn; 3China Athletics College, Beijing Sport University, Beijing 100061, China; 4School of Business, Hunan University of Science and Technology, Xiangtan 411201, China; 5Department of Medical Bioinformatics, University of Göttingen, 37077 Göttingen, Germany

**Keywords:** physical activity, interpersonal competence, quality of life, survival beliefs, interpersonal relations theory, cancer

## Abstract

This study aimed to examine the relationship between physical activity and cancer patients’ survival beliefs and constructed a mediation model involving the mediating effects of interpersonal competence and quality of life. We conducted 252 questionnaire surveys on multiple chat groups for cancer patients using the WeChat software, and assessed physical activity, survival beliefs, interpersonal competence, and quality of life using standard scales. Data were analyzed using SPSS and AMOS. There were positive correlations between physical activity and quality of life (*β* = 0.393, *p* < 0.001), physical activity and interpersonal competence (*β* = 0.385, *p* < 0.001), interpersonal competence and quality of life (*β* = 0.455, *p* < 0.001), and quality of life and survival beliefs (*β* = 0.478, *p* < 0.001). In addition, a significant mediating effect between physical activity and survival beliefs was observed between interpersonal competence and quality of life (standardized indirect effect = 0.384, *p* < 0.001). The study revealed that effective physical activity led to higher interpersonal competence, more excellent quality of life, and improved survival beliefs in cancer patients, and that the association of physical activity with improved survival beliefs was fully mediated through interpersonal competence and quality of life. The findings suggest that the relevant government should increase policy support and publicity to improve cancer patients’ participation in physical activity.

## 1. Introduction

Cancer has always been a significant threat to human health [1]. In recent decades, with breakthroughs in cancer research and advances in medical science and technology, cancer patients’ recovery and survival rates have gradually increased [2,3]. However, medical advances have not necessarily been accompanied by changes in cancer patients’ beliefs about survival. Cancer treatment is a painful process, and the effects of treatments can vary, depending on a patient’s mood. Negative emotions, such as fear, anxiety, and depression, can accelerate the deterioration caused by the disease and adversely affect treatments [4,5]. These emotions also interfere with the treatment processes followed by medical staff [6]. Therefore, maintaining a positive and optimistic mood will help alleviate patients’ fears [7]. Studies have shown that people with high survival beliefs are more active in receiving treatments [8], tend to be more self-motivated in the face of cancer stress, and face their future lives with optimism. In contrast, people with low survival beliefs tend to equate cancer with death. They are more inclined to adopt negative ways of yielding and avoiding it, thus allowing the disease to continue to develop. Therefore, we need to pay attention to the survival beliefs of cancer patients.

In contemporary medicine, previous studies have focused on the effects of treatments and medical drug interventions used on cancer, such as surgical treatments [9], radiotherapy [10], chemotherapy [11], immunotherapy [12,13], and hormone therapy [14]. However, cancer patients suffer from a decline in their physical functions and the double blow of impaired social and psychological processes. Therefore, it is not enough to evaluate the effects of treatment on a patient based on that patient’s physical state. Studies have shown that the psychological trauma left after cancer treatment, resulting in depression, anxiety, and other psychological problems, may be one of the causes of the deterioration of cancer treatments and recovery rates [15]. Therefore, psychotherapy is essential for the treatment of cancer patients. Many scholars have focused on the intervening psychological factors in cancer patients such as depression and anxiety. They have also begun to pay attention to postoperative psychotherapy interventions such as cognitive behavioral therapy [16] and musical therapy [17].

Physical activity is a simple yet effective intervention that can yield numerous benefits for individuals, including improvements in physical health, mental health, social skills, well-being, and cognitive function. For instance, physical activity can serve as a unique group communication activity, allowing individuals to connect and communicate with each other. By engaging in physical activity, patients can alleviate pain, reduce loneliness, and develop a sense of social interaction. Research has shown that physical activity also promotes interpersonal communication. Moreover, being an activity rather than a therapeutic behavior, physical activity can alleviate the treatment burden and mental stress experienced by cancer patients, leading to greater participation and improved health benefits.

Unlike previous studies, this study focused on cancer patients’ most relevant survival beliefs, as these beliefs directly impact cancer treatments. This study was also the first to focus on how physical activity could improve cancer patients’ survival beliefs. This study filled the existing research gap by linking physiological activities with psychological activities and, for the first time, quantitatively demonstrating the relationships between physical activity, interpersonal competence, quality of life, and survival beliefs. The objectives of this study were as follows: (1) to understand the survival beliefs of cancer patients, (2) to explore the factors that affected cancer patients’ survival beliefs, and (3) to report existing problems to the government and to hospitals and make appropriate suggestions.

This study used interpersonal relations theory to improve our ability to explain and predict survival beliefs. We used physical activity as an independent variable, and further examined how survival beliefs develop through the mediating roles of interpersonal competence and quality of life. The innovations of this model are as follows. Firstly, the existing research has made various innovations in the treatment and intervention methods used on cancer patients. While each treatment method has its advantages and disadvantages, it also has limitations. Cancer patients are complete living bodies, and their diseases may involve physiological and psychological factors. Often, a patient’s treatment cannot be achieved by using a singular method, and may instead require a combination of various treatment concepts and techniques. This study focused on the co-promoting effect of physical activity on cancer patients’ physical and psychological characteristics. For cancer patients, physical activity can decrease pain, relieve fatigue, improve physical performance, and reduce anxiety, stress, and depression. Secondly, previous research, which had emphasized survival, had been limited to treating cancer patients through surgery or psychological interventions, and had rarely involved lifestyle and other aspects. This study found that interpersonal competence has an important impact on the survival beliefs of cancer patients. The patients in the study talked to each other and communicated emotionally when interacting with others. This behavior can promote psychological satisfaction and enhance mutual belief support, which are conducive to remission and recovery from the disease.

## 2. Literature Interview and Hypothesis Development

### 2.1. Physical Activity, Interpersonal Competence, and Quality of Life

Physical activity refers to all physical movements that consume energy due to skeletal muscle contraction, including sports, entertainment, housework, and other activities [18]. Physical activity is a unique social activity with fitness, recreational, and social functions. Firstly, the essential function of sports is the strengthening of the body, and physical activities can improve human operation [19]. Secondly, physical activity has a positive regulatory effect on emotions [20]. Physical activity can produce rich emotional experiences; participation in sporting and other physical activities can enable people to digest and offset negative emotions through catharsis. Thirdly, physical activity promotes self-efficacy [21]. By completing activities or winning in sports, people experience success, enhance their self-efficacy, and develop a firm will and solid psychological endurance through continuous effort.

Interpersonal communication is one of the essential contents of people’s social lives [22]. Self-development, psychological adjustment, information transmission, the communication of needs, and relationship coordination are all inseparable from interpersonal communication. Studies have shown that physical activity is an effective way of establishing harmonious interpersonal relationships [23]. Participation in sports activities can help people who have never met before get to know each other. It can also make patients more willing to socialize and make new friends and thus improve their interpersonal competence [24,25]. Good interpersonal competence can relieve a depressed mood, release pressure, help improve quality of life [25], and enhance anticancer enthusiasm in cancer patients.

Quality of life is a relatively broad concept that is mainly affected by the complex influences of an individual’s physical health, mental state, degree of independence, social relations, and personal beliefs and prominent features of the environment [26]. Past research has shown that physical activity benefits cancer patients [27] by improving bodily functions and quality of life [28], during both cancer treatment and recovery [29,30]. In addition, some data suggest that exercise can improve sleep quality in patients [31]. Physical activity can also provide significant psychological and cognitive benefits [32,33]. As an effective means of relieving psychological pressures, physical activity can induce positive thinking and emotions and help a patient resist negative emotions such as anxiety and tension.

In summary, physical activity not only improves the body’s overall function and enhances physical fitness but also promotes the development of an individual’s mental health and improves their quality of life. At the same time, it provides opportunities for interpersonal communication, promotes mutual communication between individuals, and facilitates people’s social adaptation, thereby indirectly improving their living conditions. Therefore, this study proposes the following three hypotheses:

**Hypothesis** **1** **(H1):**
*Physical activity positively affects interpersonal competence.*


**Hypothesis** **2** **(H2):**
*Physical activity positively affects quality of life.*


**Hypothesis** **3** **(H3):**
*Interpersonal competence mediates the relationship between physical activity and quality of life.*


### 2.2. The Mediating Effect

Survival belief is the fundamental belief in the value of life. It is an important spiritual activity in human beings and a powerful driving force for people to firmly pursue the ideal of continued survival, especially when confronted by particular negative circumstances and unfavorable situations [34]. A belief in improvement guides people’s practical lives and has instructive and normative significance [35]. Survival belief comprises three factors: a rational cognitive factor, an emotional aspect, and a deliberate element [36]. People, in their long-term life practices, gradually form survival beliefs. In addition to their subjective conditions, beliefs also include the long-term impacts of their social environments [34]. Positive survival beliefs often go hand in hand with healthy mindsets. Relevant studies have shown that individuals with higher survival beliefs are less likely to be at risk of depression, hopelessness, and suicide [37]. Individuals with solid survival beliefs will show optimism, high self-esteem, a strong faith in the meaning of life, gratitude, and other characteristics.

Good interpersonal relationships are essential parts of life. Through contact and communication, people can exchange information, broaden their horizons, enhance their emotions, maintain good attitudes, and realize the value of life. Interpersonal communication is accompanied by cognitive, emotional, and volitional factors that seem highly related to survival belief. Studies have shown that social practice activities, family support, and peer support can improve individuals’ survival beliefs [38,39]. Quality of life is a multidimensional structure that includes mental, physical, economic, social, and spiritual health [40,41]. As a comprehensive health evaluation, quality of life reflects subjective conditions, the pros and cons of the social environment, and the personal ability to adapt to society. People with a high quality of life have suitable living environments, good tolerance, and relatively high survival beliefs [42].

Relevant studies have proven that physical activity promotes opportunities for individuals to communicate with each other. There are also significant relationships between physical activity, interpersonal competence, and quality of life. In addition, research has revealed a strong association between interpersonal competence, quality of life, and beliefs regarding survival. Does physical activity impact survival belief by mediating interpersonal competence and quality of life? Based on this question, this study proposes Hypothesis 4:

**Hypothesis** **4** **(H4):**
*Interpersonal competence and quality of life mediate the relationship between physical activity and survival beliefs.*


The hypothesized model is shown in Figure 1.

## 3. Methods

### 3.1. Participants and Procedures

In this study, a snowball sampling method was used to sample cancer patients. Questionnaires were distributed to multiple chat groups for cancer patients through WeChat. Before completing the questionnaire, participants were told the following:Participation by filling out the questionnaire was voluntary;If they felt uncomfortable during the questionnaire filling process, they could refuse to continue at any time;The questionnaire was anonymous;The data analysis would be used only for academic research;Personal information and data would not be shared with anyone.

Questionnaire respondents were only allowed to fill in the formal questionnaire after reading and accepting the contents of the informed consent form. After completing the questionnaire, respondents could choose to receive electronic shopping vouchers as rewards for participation. Only one submission was allowed per person. Data were collected continuously from March to May 2022 after the Ethics Committee approved the study. In this survey, a total of 278 online questionnaires were collected and subsequently screened for answer regularity and short response time. After this screening process, 26 invalid questionnaires were excluded, resulting in a final sample size of 252 valid questionnaires. The effective questionnaire recovery rate, which represented the proportion of valid questionnaires retrieved out of the total number distributed, was calculated to be 90.6%. Considering the specific cultural background of China, the cancer patients were not asked to provide too much personal background information; otherwise, receiving a valid questionnaire would have been difficult. Table 1 shows the simple demographics of the patients. The sample consisted of 55% males and 45% females, aged 23 to 78 years, with a mean age of 57.

### 3.2. Measures

In this study, measured responses belonged to one of four categories: physical activity, interpersonal competence, quality of life, and survival beliefs. In order to adapt to the research field and specific cultural background, remove any language barriers, and generate informed feedback, the researchers made certain adjustments to the items of the scale. The scale translation process was based on Brislin’s translation model [43], which encompasses three key stages: translation, back-translation, and cultural adjustment. Firstly, two medical experts who were native Chinese speakers independently translated the scale. Next, a medical specialist was invited to review and refine the differences between the two translations to generate the first draft of the Chinese version of the scale. Secondly, a doctor of psychology with overseas study experience back-translated the first draft of the scale and compared the translated English version to the original English entries. This comparison helped identify and correct any linguistic inaccuracies in the translated text. Finally, an expert committee comprising members of the research group and the translation team compared and discussed the original scale, translated version, and back-translated version. The committee made adjustments and modifications to the translated version of the scale to ensure its accuracy, cultural relevance, and suitability for use in the Chinese population. This rigorous process culminated in the development of the final Chinese version of the test scale. In addition, a pilot test was conducted to verify the reliability of the adjusted test scale [44]. In the pilot test, 60 questionnaires were distributed using random sampling and a total of 55 valid questionnaires were returned. The Cronbach’s alpha coefficients were all higher than 0.7, indicating that the measurement tools had good internal consistency [45].

#### 3.2.1. Physical Activity

To measure the effectiveness of physical activity, this study extracted three items from the Physical Activity Scale developed by Andersen et al. [46]. An example item includes “In your leisure time, how often do you engage in light physical activity such as walking, light cleaning, raking lawn, or lightly strenuous exercise such as yoga, bowling or similar activities?” Respondents answered using a five-point Likert scale (1 = “never”, 5 = “very frequently”). The Cronbach’s alpha coefficient for this scale was 0.791.

#### 3.2.2. Interpersonal Competence

The measure of interpersonal competence was taken from the Interpersonal Competence Scale developed by Buhrmester et al. [47]. An example item from this scale is “Finding and suggesting things to do with new people whom you find interesting and attractive.” Respondents rated how much they agreed with the statement using a five-point Likert scale (1 = “strongly disagree”, 5 = “strongly agree”). The Cronbach’s alpha coefficient for this scale was 0.799.

#### 3.2.3. Quality of Life

The measure of quality of life was derived from the WHOQOL-BREF scale developed by the WHOQOL group [48]. An example item from this scale is “Do you have enough energy for everyday life?” Respondents rated how much they agreed with the statement using a five-point Likert scale (1 = “strongly disagree”, 5 = “strongly agree”). The Cronbach’s alpha coefficient for this scale was 0.831.

#### 3.2.4. Survival Beliefs

Survival beliefs were measured by using four items from the Survival and Coping Beliefs scale developed by Linehan et al. [49]. An example item from this scale is “I have the courage to face life.” Respondents rated how much they agreed with the statement using a five-point Likert scale (1 = “strongly disagree”, 5 = “strongly agree”). The Cronbach’s alpha coefficient for this scale was 0.910.

### 3.3. Data Analysis

In this study, a structural equation model (SEM) was constructed using AMOS v.26 and the model parameters were estimated using the maximum likelihood (ML) estimation method. SEM is often used to evaluate latent variables in measurement models and to test hypotheses between latent variables in structural models [50]. To evaluate both measurement and structural models, this study utilized a two-step modeling approach. [51]. First of all, we evaluated the reliability and validity of our model. Fornell and Larcker suggested that reliability analysis should be validated by using Cronbach’s alpha coefficient and the CR coefficient of latent variables [45]. A high reliability coefficient indicates that the results of a test are more consistent, stable, and reliable. Hair et al. recommend that if both the Cronbach’s alpha coefficient and CR coefficient are greater than 0.7, the test can be considered reliable [52]. Convergence effectiveness is usually measured by factor loading the coefficient and average variance extracted (AVE). Fornell and Larcker recommended that the ideal standard value of AVE must be greater than 0.5 [45]. We also evaluated discriminant ability, which requires that the square root of the AVE value of the variable be greater than the correlation coefficient between the variables. In addition, the fit and path coefficients of the hypothetical model were measured, and the existence of the mediation effect was tested. Hair et al. [52] proposed that the basic fitness of the entire model could not have a negative error or residual terms and that otherwise the test criteria would be violated. More importantly, this study followed the suggestion of Bollen and Stine [53] by using the bootstrap method to verify the existence of the mediating effect.

Considering that all data were generated by cancer patients’ self-reports, common method variance (CMV) was deemed likely to exist. Harman’s single factor test was first used to check the data. To determine the number of factors required to account for the variance of each variable, the unrotated factor solution was examined in a one-way test [54]. All the variables were incorporated into a factor analysis in SPSS, and the extraction factor was set to 1. The results showed that the cumulative variance was 44.01%, which was below the critical criterion of 50% [55]. This means that there was no common methodological bias in this study [45]. In accordance with Mossholder et al.’s [56] recommendations, this study compared the degrees of freedom and chi-square values of model 1 and model 2 to assess the fit of the models. Model 1 had a chi-square value of 614.3 with 77 degrees of freedom and a *p*-value of less than 0.001, while model 2 had a chi-square value of 195.7 with 71 degrees of freedom and a *p*-value of less than 0.05. These results indicated that the fit of model 1 was proportional to that of model 2. As there was no evidence of univariate structure, the impact of common method variance (CMV) on the study was deemed to be small and negligible.

## 4. Results

### 4.1. Measurement Model

Non-external consistency describes the reliability of a structure. The reliability test results in Table 2 show that the Cronbach’s alpha coefficient of each variable was between 0.791 and 0.910, which was higher than the recommended value of 0.7. The CR coefficients for each variable ranged from 0.795 to 0.914: much higher than the required value of 0.7. Therefore, these variables had good reliability. The further confirmation of convergent validity and discriminative validity is crucial for assessing the authenticity of scientific data. As shown in Table 2, the AVE of each variable was between 0.526 and 0.728, exceeding the recommended value of 0.5, and the factor loading coefficient of each measurement item was between 0.640 and 0.918. The research results in Table 3 all met the requirement of discriminant validity. The chosen variables all had good convergent validity and discriminative validity.

### 4.2. Structural Path Model

The error term and residual term of the structural equation model did not exhibit negative values. The structural model also showed a good fit to the data (χ^2^/df = 1.336, GFI = 0.952, NFI = 0.950, CFI = 0.987, TLI = 0.983, RMSEA = 0.037). Table 2 lists the correlations between the variables. Significant positive correlations were found between independent variables, mediators, and dependent variables, providing preliminary support for the validation of the research hypothesis. The structural pathway model results are shown in Figure 2. The conceptual model suggests that physical activity positively affects survival beliefs through two mediating factors, namely interpersonal competence and quality of life.

The effect of physical activity on interpersonal competence was statistically significant (*β* = 0.385, *p* < 0.001), supporting H1. The results showed that physical activity had a statistically significant effect on quality of life (*β* = 0.393, *p* < 0.001), supporting H2. Table 4 presents the results for the 5000 bootstrapped samples, with 95% confidence intervals. All Z values in the table are greater than 1.96, and there are no zeros within the 95% confidence intervals. In addition, the study showed that there was a possible significant mediating effect between physical activity and quality of life through interpersonal competence (standardized indirect effect = 0.175, *p* < 0.01), providing support for H3. The study also showed that there was a possible significant mediating effect between physical activity and survival beliefs through interpersonal competence and quality of life (standardized indirect effect = 0.384, *p* < 0.001), providing support for H4. These findings imply that people with high physical activity, good relationships, and a high quality of life are more likely to have higher survival beliefs.

## 5. Discussion

### 5.1. Contributions

This study represents the first attempt to utilize interpersonal relationship theory to elucidate the effects of physical activity on interpersonal competence, quality of life, and survival beliefs. This novel research perspective provides a valuable direction for understanding the mind–body connection in cancer patients. Moreover, the findings of this study have the potential to enrich the relevant viewpoints and research fields in interpersonal relationship theory.

Firstly, this study introduced a research model that illuminated the impact of physical activity on both interpersonal competence and quality of life, and found that physical activity significantly and positively impacted interpersonal competence and quality of life, consistent with previous research studies [24,57,58,59]. The effects of physical activity on improving human functions, improving sleep quality, reducing depression levels, and promoting interpersonal communication have been verified. Physical activity is a process in which individuals actively participate, and is not limited to sporting events. At the same time, participation provides methods for individuals to improve their ability to withstand stress in life, relieve interpersonal tension, and effectively improve their mental health [60,61]. Appropriate physical activity can also activate people’s cognition and behavior, promote their psychological balance, and strengthen their self-awareness and positive emotions [62] to help them maintain a higher level of quality of life.

Secondly, the research model confirms that interpersonal competence serves as a mediator between physical activity and quality of life. According to the theory of interpersonal relations, the formation process of interpersonal relationships must go through four stages: recognition, confirmation, progression, and resolution [63]. Physical activities are always carried out in a particular social environment that requires interactions and connections between people. With changes in the contents of these activities, it may be necessary for people to assume different roles, becoming organizers, practitioners, and helpers. Therefore, in sports activities, one must correctly understand one’s position and undertake the corresponding responsibilities and obligations to coordinate all aspects of their relationships; this is essential to improve interpersonal competence. Through cooperation and competition, people realize this communication and promote the establishment and harmonious development of interpersonal relationships [25]. In contrast, the lack of interpersonal competence will lead to interpersonal disorders, which are some of the causes of psychological problems. Mental health problems in interpersonal relationships can manifest as egotism, suspiciousness, shyness, withdrawal, inferiority complex, jealousy, and social fear. These psychological barriers will significantly affect people’s quality of life. They can be contrasted with good interpersonal competence, which helps people maintain good attitudes while dealing with various complex social relations, better adapt to society, and improve their quality of life [64].

Finally, our model sheds light on the positive effects of physical activity on enhancing survival beliefs in cancer patients. Furthermore, our study investigated the specific mechanism underlying this effect by incorporating interpersonal competence and quality of life as mediator variables. The findings showed that the positive effect of physical activity on survival beliefs was fully mediated by interpersonal competence and quality of life. As shown in Figure 2, interpersonal competence and quality of life can explain 48% of the variation in survival beliefs, which is significantly higher than the general reference value. As a basic form of spiritual consciousness, survival belief is not fixed but will change with objective reality and can be constantly adjusted and improved. Interpersonal competence can help individuals cope with emergencies in life and work through family or peer support [65]. When cancer patients have high interpersonal competence, they can analyze the impact of crises on themselves and use interpersonal relationships to regulate their emotional states and behaviors [66]. As a result, they can put enough energy into their behavior, actively mobilize their ability to deal with problems, establish healthy lives, have more positive emotions, and maintain a good quality of life. At the same time, the higher the quality of life of cancer patients is, the more they can reflect on their advantages in the living environment, and the higher their psychological, physiological, and spiritual health levels are, the more they can have a positive sense of their selves, love lives, and life in general, which will then enhance survival beliefs. These findings confirm the effectiveness of physical activity in improving one’s physiological and psychological health, affecting interpersonal competence and quality of life and thereby indirectly affecting the survival beliefs of cancer patients and improving their survival.

### 5.2. Implications

This study demonstrates the positive impact of physical activity on survival beliefs in cancer patients and the mechanism of action during this effect. Interpersonal competence and quality of life can directly affect survival beliefs, while physical activity can directly affect interpersonal competence and quality of life. Therefore, simply by increasing their participation in physical activity, cancer patients can indirectly and positively impact their beliefs about survival. Physical activity is an essential avenue that people of all ages and abilities can partake in for cancer prevention, treatment, and control [67]. As an effective cancer treatment intervention, physical activity has the following advantages: it helps improve the quality of life of patients, is easy to implement, has no treatment costs, and does not need to target specific cancer patients.

Considering the positive impact of physical activity on interpersonal competence, quality of life, and survival beliefs, the government should increase policy support for sports activities, encourage the active participation of patients, and promote the formation of lifelong sports awareness. To increase physical activity, the government should increase the relevant financial investment. Such investment could support hospitals in sports training and organization management and could also be used to construct sports venues and diversified sports facilities in hospitals. In addition, hospitals should establish sports rehabilitation management departments and train professional sports instructors. Finally, relevant lectures on physical fitness should be held to publicize the importance of sports and physical activity to allow more patients to participate in sports interactions. Medical staff should customize an individualized physical activity plan for each patient willing to engage in physical activities. Then, the team should reasonably guide patients on the correct way to engage in, and content of, each physical activity. It can also be appropriate for the team to participate in activities with patients to improve the patients’ goodwill towards medical staff.

### 5.3. Limitations of the Study

This study has certain limitations. First, the sampling procedure for this study was not pure randomization; this limits the generalizability of its findings. Future studies should overcome this limitation and use purely random sampling to make the results generalizable. Second, this study was cross-sectional and only considers a superficial level of data. Longitudinal studies should be considered for future research to make studies more in-depth and meaningful. Third, this study only performed a simple pooled analysis without distinguishing the characteristics of disease in the study samples. Future research may consider a classification analysis to examine differences between different conditions.

## 6. Conclusions

The study found three critical cores that stood out among the cancer patients. First, cancer patients with higher physical activity had higher survival beliefs. Second, interpersonal competence and quality of life were related to survival beliefs in cancer patients. Finally, physical activity affected survival beliefs by mediating interpersonal competence and quality of life. This study and similar studies suggest that if cancer patients with low survival beliefs can improve their survival beliefs and be willing to participate in physical activity, treatments to increase survival beliefs and prolong survival may be enhanced. Consequently, this study suggests that the government should improve the corresponding policies, while hospitals should reinforce the promotion of lifelong sports awareness to heighten participation in physical activity among cancer patients.

## Figures and Tables

**Figure 1 behavsci-13-00449-f001:**
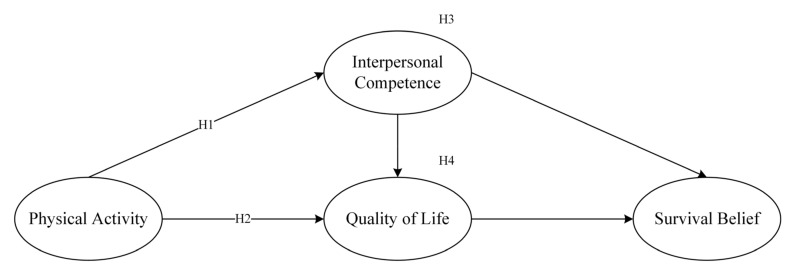
The hypothesized model.

**Figure 2 behavsci-13-00449-f002:**
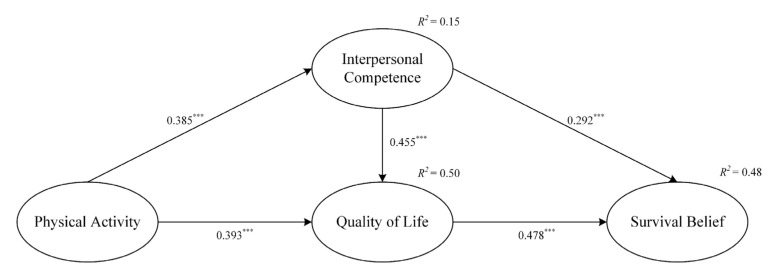
Structural path model. *** *p* < 0.001; standardized coefficients are reported.

**Table 1 behavsci-13-00449-t001:** Participant profile (N = 252).

Variables		N	%
Gender	Male	139	55.2
	Female	113	44.8
Age (years)	Mean: 57		
	Range: 23–78		
	<60	130	51.6
	≥60	122	48.4

**Table 2 behavsci-13-00449-t002:** Reliability and validity tests.

Items	Loadings	Cα	AVE	CR
Physical Activity (PA)		0.791	0.564	0.795
PA1	0.770			
PA2	0.756			
PA 3	0.726			
Interpersonal Competence (IC)		0.799	0.526	0.814
IC1	0.791			
IC2	0.640			
IC3	0.801			
IC4	0.653			
Quality of Life (QoL)		0.831	0.626	0.833
QoL1	0.780			
QoL2	0.847			
QoL3	0.742			
Survival Beliefs (SB)		0.910	0.728	0.914
SB1	0.869			
SB2	0.859			
SB3	0.918			
SB4	0.759			

All standardized loadings were significant at the 0.001 level.

**Table 3 behavsci-13-00449-t003:** Discriminant validity test.

Construct	PA	IC	QoL	SB
PA	**(0.751)**			
IC	0.310 **	**(0.725)**		
QoL	0.473 **	0.499 **	**(0.791)**	
SB	0.368 **	0.520 **	0.590 **	**(0.853)**

The square root of the average variance extracted (AVE) is shown in diagonals (bold); the off-diagonals are Pearson’s correlations of constructs. ** *p* < 0.01.

**Table 4 behavsci-13-00449-t004:** Standardized indirect effects.

	Point Estimate	Product of Coefficients		Bootstrapping				
				Percentile 95% CI		Bias-Corrected 95% CI		Two-Tailed Significance
		SE	Z	Lower	Upper	Lower	Upper	
Indirect effects								
PA→ QoL	0.175	0.055	3.182	0.079	0.289	0.081	0.293	0.001(**)
PA → SB	0.384	0.066	5.818	0.259	0.520	0.259	0.518	0.000(***)
IC → SB	0.218	0.088	2.477	0.083	0.425	0.081	0.416	0.013(*)

Standardized estimation of 5000 bootstrap samples; * *p* < 0.05, ** *p* < 0.01, *** *p* < 0.001.

## Data Availability

The data used to support the findings of this study are available from the corresponding author upon request.

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
