# Peer review of "Active for Life after Cancer: Association of Physical Activity with Cancer Patients’ Interpersonal Competence, Quality of Life, and Survival Beliefs"

_behavsci, 2023, doi:10.3390/bs13060449_

Round 1

Reviewer 1 Report

Dear authors,

About your paper titled: Active for life after cancer: Effect of physical activity on cancer patients’ interpersonal competence, quality of life and survival beliefs

Follow my comments and suggestions

Title

Replace “effects” with “relationship” or “association” because, in actuality, you performed correlational instead of experimental statistics.

Abstract

The same case is repeated in the abstract when you cite the effect of physical activity. Replace it by association or keep using the term mediating.

All abstract text is written in passive subject. Please, consider rearrange to active subject as many times as possible in the text structure.

Keywords

You repeated the same keywords present in the title and abstract text. You need to replace the keywords with synonyms from the medical and health Thesaurus.

Introduction

Lack of standardization in the apostrophe use of the example “patient’s” in sometimes and in others passages “patients’.”

In the second paragraph, you described a lot of treatments but not physical exercise. Lines 88-93 must be rearranged right after the second paragraph (line 58), creating a link between the importance of physical exercise before you expose the research gap.

Further, you use verbs in the infinitive form in several parts of the introduction. Avoid this verbal usage, and replace it with an active subject in the present tense as much as possible.

2.2. The mediating role of interpersonal competence and quality of life

There are repetitive words, for example:  “Belief is the binding force that supports people in persevering. It guides people’s practical life with the belief in the upward value and has instructive and normative significance”

Please review this session, aiming to shortly as many as possible the number of words used. I advise you to join short sentences in only one sentence, reducing the use of repetitive words.

Figure 1

The figure is cut by the figure not. Please, readjust this figure.

3.3. Data Analysis

The data analysis description is very superficial. Please, report more mathematical details in these procedures.

4. Results

4.1. Measurement Model

In lines [(252 to 256), (263-266), (278-279), (291-293)], your reports are repetitive and could be presented in the statistical procedures. All of this information must be reported in statistical procedures or methodology (where you judge more conveniently) but not in results. Avoid discussing subjects in results and look to keep only numerical reports.

6. Conclusions

Remove the following sentence in the first line of results: “This study explores the impact mechanism of physical activity on the survival beliefs of cancer patients from the perspective of interpersonal relations.”

Correct “There-fore” by “Therefore” in line 438.

The sentence in lines 439-443 is very complicated. Simplify this description without losing the essential information.

No further comments.

Best regards.

Dear editors of the Behavioral Sciences,

After analysis of the paper titled: Active for life after cancer: Effect of physical activity on cancer patients’ interpersonal competence, quality of life and survival beliefs

This paper attends the needed quality to be published in the Behavioral Sciences.

No further comments,

Best regards.

Reviewer 2 Report

1.      I am interested in understanding the rationale and justification for including a regression path between mediators in your proposed model (H4). How does this regression path align with your research question and theoretical framework? I would like to seek further clarification regarding the inclusion of a regression path between mediators, as this is not a standard procedure in SEM (Hair et al., 2019; Hayes, 2018).

2.      I understand that you faced challenges in discriminating between participants who had cancer, were currently patients, and those who had never had cancer. It would be helpful if you could provide additional details on how you mitigated this issue during the data collection process. Did you use a screening questionnaire or any other approach to minimize potential biases? How did you ensure that the data obtained were reliable and accurately reflected the intended study population?

3.      I would like to kindly request that you provide a clear and transparent explanation for the absence of sample size calculations in your study. Sample size calculations are crucial in determining the appropriate number of participants needed to detect statistically significant effects with adequate power, and they play a crucial role in ensuring the validity and reliability of study findings.

4.      I would like to request additional information regarding the sociodemographic characteristics of your sample in your manuscript. I believe that reporting sociodemographic data is essential in understanding the context and generalizability of your findings. Please provide more comprehensive information about the sociodemographic characteristics of your sample, including relevant demographic factors such as age, gender, race/ethnicity, socioeconomic status, education level, and other relevant sociodemographic variables that may impact the interpretation and generalization of your results such as cancer type, cancer stage, treatment status, duration of illness, and any other relevant clinical details that are pertinent to your research question.

5.      I would like to bring to the attention of the authors the lack of acknowledgment regarding the use of English validated measures in a sample of Chinese students in the manuscript. The authors have mentioned the use of validated measures in their study, but there is no mention of the process of translation and adaptation of these measures from English to Chinese. It is crucial to acknowledge and describe the process of translation and adaptation of measures to ensure that the validity and reliability of the measures are maintained in the target cultural context.

6.      I am interested in understanding the rationale and justification for aggregating the first-order factors of the WHOQOL into a single global factor. It would be helpful if you provide information on how the aggregation was performed, including any statistical or psychometric techniques used, and the rationale for the specific aggregation method chosen.

7.      Line 237: Again, would like to raise a concern regarding the proposed model that includes a regression path between mediators, which is not a standard procedure in SEM. Additionally, it appears that the proposed model may not be compatible with IBM SPSS AMOS, a popular software for conducting SEM analysis.

8.      I would like to emphasize the need for caution in interpreting the results as indicating a definitive mediation effect. Based on the results presented in your manuscript, it is important to note that the performed analyses do not provide conclusive evidence of a mediation effect, but rather suggest a possible mediating effect based on the indirect effect estimated in the SEM.

9.      I have reviewed your manuscript and noticed that the AVE results are reported differently in two different sections of the manuscript (table 2 and 3). As a reviewer, I would like to request clarification and consistency in reporting the AVE results throughout the manuscript. Please provide an explanation for the discrepancy in the AVE results reported in the two sections and ensure that the reported results are consistent and aligned.

10.   Additionally, I would like to request that you disclose the specific cutoffs used for AVE scores and internal reliability coefficients, as these are important information for evaluating the quality and reliability of your measurement model. Including this information will enhance the transparency and rigor of your research.

11.   I would like to indicate that I will not provide specific comments on the discussion section of the study at this time, as it is likely to undergo substantial changes based on my comments and feedback. The discussion section is typically the part of the manuscript that reflects on the interpretation of the results and provides insights, implications, and future directions based on the findings. However, as my comments on other sections of the manuscript are intended to address several issues and may impact the interpretation of the results, it is possible that the discussion section will need to be revised accordingly to ensure consistency and coherence throughout the manuscript.

Needs revisions.

Reviewer 3 Report

Dear authors, I would like to congratulate you for the article.

Very interesting, well written and organized.

I was expecting to see more information about the participants in the description of the sample, although the information they present is also sufficient.

Kind regards.

Round 2

Reviewer 1 Report

Dear authors, follow my last reviews in the cover letter. 

After to perform the last modification, your paper will be ready to be published in Behavioral Sciences (MDPI)

Best regards

Author Response

Dear reviewer,

Thank you for addressing the issues mentioned earlier. As non-native English speakers, our grammar revisions may not have met your expectations. We apologize for this. To ensure the necessary improvements, we have opted to utilize the paid English editing service offered by MDPI Journal. We appreciate the time you've dedicated to reviewing our manuscript, as it has been immensely valuable to us.

Reviewer 2 Report

The authors did a good job reviewing their Manuscript. 

Grammar and ponctuation review.

Author Response

(The authors gave the same response as above.)
